# Nurses' perspectives on the provision of self-care management education to patients with heart failure: A qualitative study at Cardiac Hospital, Dar es Salaam Tanzania

**Peter M. Shirima** [1,2]*, **Dickson A. Mkoka**[2], **Masunga K. Iseselo**[2]

**1** Department of Nursing Services, Muhimbili National Hospital-Mloganzila, Dar es Salaam, Tanzania,
**2** Department of Clinical Nursing, Muhimbili University of Health and Allied Sciences, Dar es Salaam, Tanzania

* pkasongoro@gmail.com

**Data Availability Statement:** All relevant data are within the paper and its Supporting Information files.

## Abstract

### Background

Patients' education along with a motivation for developing self-care management skills is an essential component in the management of heart failure(HF). Self-care management education has been practiced by nurses in many hospitals. However, there is inadequate evidence for the provision of self-care management education in low-income countries including Tanzania. Lack of self-care management education to patients with HF during discharge is the most common reason for re-admission to hospitals.

### Aim

This study aimed to explore nurses' perspectives focusing on facilitators and barriers to the provision of self-care management education to patients with heart failure at Jakaya Kikwete Cardiac Institute, Dar es Salaam, Tanzania.

### Materials and methods

This study used a qualitative descriptive design. Purposive sampling was used to recruit 12 study participants. In-depth interviews were used to collect the data. We used thematic analysis to come up with the themes and sub-themes.

### Results

The two major themes emerged from this study; The first theme is "Improved patient quality of life and health outcome" which describes factors that motivate nurses to continue giving self-care management education to heart failure patients. The second theme is "Reduced effective uptake of self-care management education" which describes nurses'perspectives on barriers for providing self-care management education to heart falure patients. Nurses highlight some barriers while providing self-care management education to patients with heart failure including;inadequate knowledge of self-care management among nurses, lack

**Funding:** This research receive limited fund support from Muhimbili University of Health and Allied Sciences under a Project named East Africa Centre of excellence for Cardiovascular Sciences.

**Competing interests:** The authors have declared that no competing interests exist.

of privacy during the provision of self-care education, poor communication skills among nurses, and lack of learning materials. Also, nurses pointed out facilitators that influence the provision of self-care management education such as increased involvement of family members and the use of peer educators.

## Conclusions and recommendations

Poor self-care management for patients with heart failure results in readmission and prolonged hospital stay. Family involvement and the use of peer educators are the key steps in the improvement of self-care management for patients with HF. However, patient cognitive impairment and poverty which contribute to poor health outcomes, should be taken into consideration when planning for discharge for patients with HF. Self-care management education should be part of routine health care.

## Introduction

Heart Failure(HF) has been defined as a global pandemic since it affects around 26 million people worldwide [1]. Cardiovascular diseases(CVDs) alone including heart failure are responsible for 13% of the total NCD deaths in Tanzania and adults aged 25–64 years are highly affected [2]. In Tanzania, mortality rates according to age show higher rates in men compared to women (473 versus 382 per 10,000 population). Also, it shows an increase in the prevalence of CVD death rates from 9% in 2012 to 13% in 2016 [3]. The increased prevalence was reported to be driven by the growing trends of CVDs in Tanzania.

Self-care management is considered a cornerstone in managing patients with HF and contributes to the improvement of treatment effectiveness and reduces hospital admission [4]. Self-care management can be defined as a process of maintaining health through health promotion and preventive practices. Patients with HF who actively engage in effective self-care activities have better quality of life and lower mortality and readmission rates than those with less engagement in self-care [5]. According to a study reported in China, self-care consists of specific behaviours patients perform to control disease and maintain health [6]. The study elaborates that, for patients with HF, self-care includes adhering to medications and treatment, avoiding excessive fluid and salt intake, and monitoring daily weight. Also, engaging in exercise, identifying exacerbating symptoms, and taking appropriate steps to intervene if symptoms worsened [7].

Exacerbation of symptoms because of non-compliance with the therapeutic regimen as well as a lack of timely treatment seeking after the onset of decompensation, considered the most common leading cause of readmissions [8]. Hence, providing self-care management education during the discharge of patients with HF, along with a motivation for developing self-care management plans and skills is an essential component in the management. Previous studies have shown that if patients practice constant self-care, 30% of hospital admissions and more than half of the readmissions can be averted [6]. As hospital stays become shorter and less frequent, the responsibility for self-care management shifts to the patients and their families. Therefore, self-care management interventions that promote and support self-care during and after hospital discharge become valuable. Another study done in Sweden shows that lack of self-care management education for a patient with HF was the most common reason for readmission and complicated many more admissions [9]. Data from various populations suggested that up to 50% of patients hospitalized due to HF are readmitted within 6 months.

In Tanzania, there are limited empirical studies that describe self-care management education to patients with heart failure. Nurses and other health care providers in clinical settings are required to ensure effective delivery of self-care management education as part of their routine care. However, despite its patient recommendation's optimal health outcome, several studies report complicities in the delivery of self-care management education. There is scarce empirical information on what drives or hinders nurses in the effective delivery of self-care management education in the Tanzanian context. To explore this, we conducted a qualitative study aimed at understanding the nurse's perspectives on motivators and barriers to the provision of self-care management education to patients with HF in Dar es Salaam.

## Materials and methods

### Study design

A descriptive qualitative study design was used to explore nurses' views, facilitators, and barriers to the provision of self-care management education to patients with HF.

### Study areas and setting

The study was conducted at Jakaya Kikwete Cardiac Institute (JKCI) in Dar es Salaam, Tanzania. JKCI is a tertiary hospital in Tanzania. It accommodates patients with cardiovascular problems from all over the country. The Institute had a bed capacity of 120 with an average of 700 outpatients and 100 inpatients. There were about 120 nurses employed in this hospital distributed well in different units including wards, intensive care, operating theatre, interventional laboratory, and clinics and some of them are administrators.

### Study participants

This study recruited nurses working in different units in the hospitals for more than one year and had experience caring for patients with HF. We excluded the participants who were on leave because it was difficult to reach them. Also, nurses on administrative duties were excluded from the study due to a lack of day to day contact with patients. None of the participants recruited declined to participate.

### Sample size and sampling procedure

A total of twelve (12) participants were purposely selected and interviewed. Using the duty roster, the first author (PS), assisted with unit in-charge selecting nurses who had more than one year of experience and with different levels of education. Recruitment of participants for this study took one month time from 19th April to 20th May 2022. We did not predetermine the sample size, however, the interview process ended up with the 12th participant after noting repetition of information with no new insight in relation to our research objectives.

### Data collection

The in-depth interview (IDI) method was used during data collection. The first author conducted all the interviews with the participants. Before the commencement of data collection, a semi-structured interview guide was developed from the literature review [10, 11] and was reviewed by a researcher who had experience in self-care management for patients with HF. On the day of the interview, participants were visited at their units and requested to choose a convenient time. Interviews were conducted at the unit in charge offices where there was good ventilation and a calm environment for comfort, and there were no interruptions from other people. The duration of each interview was between 30 and 45 minutes. The interview focused

on the questions about patient care, family involvement, counseling style, and education materials present at the hospital, that answered the research questions first to ensure the completeness of the interview until saturation was reached. All interviews were conducted in the Swahili language (the native language of the interviewer and interviewees), and information was recorded using notebooks and an audio digital recorder after all participants gave consent to do that.

## Data analysis

The data generated from interviews were reviewed daily to ensure accuracy and completeness. Audio-recorded information was transcribed verbatim after each interview. The transcribed data were checked against audio data each day to correct omissions and errors. After the transcription, the data was translated from Swahili to English. The translation was done by the researcher and shared with the research fellow for consistency. To ensure the quality of translations, we ensure transferring the meaning of the words rather than re-writing them; so considering content equivalence in translation while maintaining semantic equivalence as is necessary to produce informative text [12, 13]. Thereafter, English transcripts were made available to the co-authors to familiarize them with the data and generate insight into the contents. The researcher analyzed the data with guidance from the co-authors using steps of thematic analysis.

The transcripts and field notes were first analyzed manually by reading and re-reading to become familiarized with the data, followed by the generation of initial codes from the transcripts. The generated codes were reviewed to identify the patterns of the data set that correspond to the research questions. Similar codes related to the present research questions were grouped to form sub-themes. Similarly, sub-themes were merged to form a theme grounded on the data [14]. The emerging themes were reviewed, defined, and then shared with other authors and discussed for their relevance to research questions. The final themes and sub-themes were reported with relevant supporting participants' quotes.

## Ethical consideration

Ethical clearance was obtained from the Research Ethics Committee (REC) of Muhimbili University of Health and Allied Sciences (MUHAS) with Ref. No. DA.282/298/01.C/1084. Permission to conduct the study was obtained from the respective hospital setting (JKCI). Written informed consent was sought from participants before they participated. They were also informed about the voluntary nature of participation and that they could withdraw from participation at any time if they wished to. Confidentiality was assured when we used participant identification numbers instead of their actual names.

## Results

### Demographic characteristics of the participants

A total of twelve (12) Registered Nurses participated in the study. The age of participants ranged from 31 and 40 years old with experience of about 5 years in managing patients with heart failure. The socio-demographic information about research participants is summarized in **Table 1**

### Themes and subthemes

Two main themes emerged from this study; The first theme is "Improved patient quality of life and health outcome" which describes factors that motivate nurses to continue giving self-care

**Table 1. Social demographic characteristics of participants.**

| Participants ID | Age (Years) | Sex | Education Level | Experience (Years) |
|---|---|---|---|---|
| P1 | 40 | Male | Degree | 6 |
| P2 | 36 | Male | Degree | 6 |
| P3 | 32 | Male | Degree | 4 |
| P4 | 33 | Male | Degree | 4 |
| P5 | 31 | Male | Degree | 4 |
| P6 | 33 | Female | Diploma | 7 |
| P7 | 32 | Female | Diploma | 7 |
| P8 | 34 | Female | Diploma | 7 |
| P9 | 33 | Female | Diploma | 4 |
| P10 | 35 | Female | Diploma | 7 |
| P11 | 32 | Female | Diploma | 4 |
| P12 | 36 | Female | Degree | 4 |

management education to heart failure patients. The second theme is "Reduced effective uptake of self-care management education" which describes nurses'perspectives on barriers for providing self-care management education to heart falure patients. The themes and corresponding subthemes that correspond to the research questions are summarized in **Table 2**.

### Theme 1: Improved patient's quality of life and health outcome

Nurses reported Improved patient quality of life after the provision of self-care management education. They explained that when patients followed instructions given by nurses during discharge, they would be free from signs and symptoms of heart failure. This would facilitate involvement in their normal daily activities.

> *"I am very convinced when I see that a patient has improved after giving him/her self-care education at home. . . when I see this improvement, I am motivated to do it for more patients when they come to get care"* (P1, BScN).

Participants also described that when patients experienced good quality care, they developed confidence and rapport with care providers. In this case, the patients felt safe and secure. Improvement demonstrated by patients following the application of advice given by nurses during hospital stays proves that patients describe good health outcomes. It encourages nurses to continue giving them self-care management education.

**Table 2. Themes and related subthemes.**

| Questions | Subthemes | Themes |
|---|---|---|
| What motivates the provision of self-care management education? | 1. Adherence to medication and lifestyle changes<br>2. Reduced readmission<br>3. Strengthened nurse-patient relationship<br>4. Involvement of family members | Improved patient quality of life and health outcome |
| What hinders the provision of self-care management education? | 1. Inadequate knowledge of self-care management<br>2. Poor communication skills<br>3. Patients cognitive impairment<br>4. Inadequate learning/teaching materials | Reduced effective uptake of self-care management education |

*"Also, for me, if I see my patients recover from illness and return to their usual activities in the community, it encourages me to provide self-care education to them..."* (P3, BScN).

Participants in this study viewed that the provision of self-care management education resulted in reduced patient readmission. According to participants, the provision of self-care education enabled patients to adhere to medications and proper diet and follow the advice given on lifestyle changes like quitting smoking, exercising, and quitting alcohol. It was also noted during the interview that, reduced readmission helps reduction of workload in the wards, and this motivates nurses to provide effective self-care management to their patients.

"*Often when we give them this kind of self-care education it helps them to get rid of dangerous things like smoking and drinking alcohol, which can cause them more problems*" (P9, DN)

*"Another issue is about self–care management after you have educated patients is reduced readmission that will automatically reduce workload for nurses... If patients come to the clinic for a check-up and go back home, the workload for nurses in the wards will decrease..."* (P1, BScN).

It was noted during the interview that family members play a very key role in helping their patients follow what had been taught during the education session. Nurses in this study described how educating patients and their families helps family members to guide their patients at home to follow what has been instructed at the hospital and hence increase education uptake. Also, several participants agreed that giving education to patients in the presence of their family members helps them to concentrate well on what has been taught. They further commented that family members may give good support to their patients after they have been involved in the process of disease management, especially at home.

*"Most of the time we involve their loved ones to make them concentrate on the health education with the people they trust most..." (P3, BScN)*

Other participants emphasized that family members play important roles in the care of patients including contributing to decision-making, assisting the healthcare team in providing care, improving patient safety and quality of care, assisting in-home care, and addressing the expectations of the patient's family and society at large.

"*We always involve their relatives, especially when the patients are not able to concentrate well in the health talk due to sickness...*" (P12, BScN).

## Theme 2: Reduced effective uptake of self-care management education

Nurses also shared their views concerning what hinders them from providing education for self-care management to patients with heart failure before their discharge. Participants viewed that, the provision of self-care management education is impaired by nurses' inadequate knowledge on self-care management, poor communication skills, and inadequate education materials.

**Nurses' inadequate knowledge of self-care management.** Inadequate knowledge of self-care management for patients with heart failure was mentioned by nurses in the unit as a prominent barrier for them to provide effective self-care education. They mentioned a lack of knowledge regarding diet and salt restriction and misconceptions about heart failure and its symptoms leading to a failure to understand the relationship between disease and symptoms.

Participants expressed that they lack knowledge regarding self-care management for patients with heart failure. They reported losing confidence while giving discharge education because they were not sure about what they were telling their patients.

*"Experts that are knowledgeable on self-care management to heart failure patients are not enough, we as a nurse we don't have enough knowledge on this issue. . . this leads to unsatisfactory transfer of this knowledge to our patients. . ." (P2, BScN)*

Lack of on-job training was mentioned by several nurses as in the following quotes;

*"There are no short courses regarding cardiovascular issues while we are in daily activities. . . most nurses do not have exposure specifically in self-care management for patients with heart failure . . ." (P2, BScN).*

Participants also pointed out that regular training would equip them with knowledge and they would gain confidence in providing effective education to the patients.

**Poor communication skills.**   Participants in this study viewed that their communication skills were not that good and impaired the provision of self-care management education to the patients during discharge as stated below;

*"Sometimes we fail to teach these patients because some of us are poor in communicating with sick patients. . ."* (P4, BScN)

Several participants agreed that the use of effective communication is an important aspect of patient care, which improves the nurse-patient relationship and has a profound effect on the patient's perceptions of healthcare quality and treatment outcomes. Further, participants emphasiezed that to effectively teach patients with heart failure, nurses must use good communication techniques including listening skills, questioning skills and interviewing skills. However, participants expressed that such techniques are not well mastered by nurses, hence limiting their role of providing effective education for self-care management. Several participants were the concern that, some nurses are not a good listener and are attentive to what patients say during provision of self-care management education. Others emphasized that, poor communication skills, not only affect the delivery and uptake of self-care manegement education but also affect the direct care provided to these patients.

**Reduced patient's cognitive impairement.**   Nurses reported that some patients are not attentive when given instructions due to decreased cognitive functions. They were of concern that for patients to take and remember the instructions given during health education, their cognitive status should be intact. They also reported that the patient's sickness, age and psychological problems might precipitate cognitive impairment to them. Nurses also stated that patients with HF demonstrate reduced global cognition as well as deficits in multiple cognitive domains including executive function, psychomotor speed, and verbal memory. Screening for cognitive impairment in HF is rarely performed in the clinical setting and cognitive dysfunction in patients with HF remains poorly recognised. Deficits in executive function in patients with HF have been associated with poor functional independence, decreased ability to manage medications as well as non-compliance to smoking cessation. All of this explains the poor health outcomes to some heart failure patients despite the effective provision of self-care management education.

"Sometimes these patients are affected on their cognitive due to sickness. . .they suffer psychotic-like syndrome. . . due to that they cannot concentrate what you are telling them. . ." (P4, BScN)

**Lack of learning/teaching materials.** It was noted that education given to patients during discharge had not been documented anywhere. Participants reported providing discharge education to the patient only by oral instructions. It was also noted that there were no fliers, posters, or brochures that were given to patients before going home.

*"There is no document like brochures or posters regarding self-care management education that has been given to patients after discharge. . ." (P12, BScN).*

Participants pointed out that the lack of fliers, posters, and brochures made nurses fail to help patients to make references while they are at home.

*"We always provide health education but not documented. . .we only instruct them what to do but we are not giving them any written document like posters for reference while at home. . ." (P4, BScN).*

Furthermore, participants mentioned that the discharge summary that is handled to patients or their relatives contains only important messages regarding what are interventions done while in the hospital, what medications had been discharged to use at home, and the return day for follow-up. It was noted that the summary does not include self-care management packages like lifestyle changes, restricted foods, psychological counseling, and how to recognize threatening events.

*"The discharge summary paper was given to patients my opinion should be updated to consider self-care package, otherwise we are not helping these patients. . ."(P1, BScN)*

## Discussion

This study explored the perspectives of nurses on the provision of self-care management education to patients with heart failure. Nurses being frontline caretakers are required to effectively engage in provision of self care management education as their key role in caring for patients with heart failure. This study found multiple factors that motivate or hinder nurses from delivering self-care education effectively to patients with heart failure. Improving patient's quality of life and optimal health outcome was found to be the main driver for nurses' active engagement in the delivery of self-care management. However, the study found low uptake of self-care management education due to nurses' inadequate knowledge about self-care management, lack of communication skills for delivering education, shortage of educative resources, and cognitive impairment of some patients with heart failures.

### Provision of self-care management education and improved patient quality of life

Despite the diversity in characteristics of nurses' clinical experience, care settings, and the study area where the study was conducted, nurses air similar views in the provision of self-care management education for heart failure patients. This study has identified the role of self-care management education provided by nurses and patients' health outcomes, low uptake of self-care management education provided to patients living with heart failure, and resource constraints for the provision of self-care management education.

Nurses reported that patients' health outcomes were facilitated by the provision of effective self-care management education. According to them, the provision of self-care management

education to patients helps them to reduce the burden of readmissions to hospitals. The same statement emphasized by the European society of Cardiology in 2021 that self-care management education given to heart failure patients during discharge help to prevent readmission rates these patients [5]. It enables them to adhere to medications and nutritional counseling and follow the advice given on lifestyle changes like quitting smoking, exercising, and quitting alcohol. All of this improves the quality of living and increases the life expectance of heart failure patients. A similar study found that as a result of reduced re-admission to the hospital, nurses consider it as the one that improves the quality of life for heart failure patients [15]. Nurses in this study reported that an adequate number of nurses in the unit to take care of this patient helped them to provide effective self-care management education to these patients. The study recommends that hospital management focus on hiring enough nurses who will join hands in the units in providing self-care management education to heart failure patients.

It was found in Singapore that, heart failure is a burden in the elderly which is related to increased hospitalization frequency, shortened life expectancy, as well as poor quality of life, and is the most prevalent heterogeneous clinical syndrome. The study proved that a program of self-care management education can be regarded as the proper method to improve the quality of life in heart failure patients [16].

Nurses reported that involving family members while providing self-care management education was the most important facilitator that encouraged them to continue giving self-care education to heart failure patients. According to Shahriari, most patients with heart failure live with other family members in their homes, and participation and support of family members can play a key role in self-care behaviors and the efficiency of disease control [17]. Similar studies revealed the association between family support and heart failure patients' self-care [18, 19]. They reported that the patients with more support had better compliance with self-care health behaviors. In this study, nurses reported that self-care management in heart failure patients was promoted after supportive intervention and the promotion of family support toward the patients. A pilot study reported that supportive intervention and conducting group sessions with heart failure patients' caregivers and counseling regarding the manner of positive and supportive communication of the family with the patients, led to better compliance concerning following a low salt diet [18]. The results of other research also showed an association between social support from the family side and self-care, as well as a positive association between family support and self-care which is consistent with the present study [19]. This study also found that reduced workload for nurses as a result of reduced re-admission of heart failure patients to hospital has been a good motivator for nurses to provide effective self-care management education to patients while they are in the hospital. However, a reduced workload might be also due to the adequate number of nurses in the unit. This study recommends that hospital management focus on hiring enough nurses who will join hands in the units in providing self-care management education to heart failure patients.

## Barriers to low uptake of self care management Education and its Implications for patient health outcome

Barriers to self-care management education as reported in this study were described by lack of knowledge among nurses, poor communication between nurses and heart failure patients, and lack of learning/teaching material. A study done in Poland [20] that shows senior nurses have low knowledge of self-care management for heart failure patients cemented the results of this current study. According to Beata, lower scores were not studied in Poland but it might be that senior nurses were less likely to attend heart failure educational sessions but they relied much on experience and previous knowledge. This study shows that discharge education was not

adequate due to a lack of knowledge and common understanding among nurses of the same unit and same hospital. Each educator instructs patients about what she/he knows about self-care management. Some of the participants mentioned poor knowledge as one of the barriers for them to provide effective self-care education to these patients.

The study done by Jeon et al in 2010 found that, to manage contextual problems such as cultural issues, health providers and educators need to have good communication skills including reflective listening, empathy, and acknowledging patients' values [21]. It was cemented by Thomas and colleagues that, effective communication skills and trust have a reciprocal relationship; by improving one the other will be strengthened [22]. However, according to our findings, poor nurse-patient communication was the most important barrier to self-care management education in patients with heart failure. This study recommends providing a safe and comfortable environment that leads to the psychological and physical comfort of the nurse and patient and facilitates using communication skills and establishing effective communication.

## Methodological consideration and study limitations

Credibility in this study was ensured through the triangulation of different study informants with various experiences who shed light on the research question from a variety of aspects. However, during both data collection and analysis, an effort was made to put the researcher's knowledge and previous experience in the management of heart failure patients in the same setting within brackets to allow the researcher to capture opinions and experiences as presented by participants [14]. Furthermore, data were analyzed and interpreted by the researcher and shared with co-authors who had adequate knowledge of the management of heart failure and qualitative studies. To enhance credibility and dependability, the data collected from interviews were triangulated with those from field notes during the analysis process, and categories and themes were shared with all authors who gave critical comments and suggestions. To confirm that the findings reflected the participants' perspectives rather than the researchers' understanding of the problem, the presented findings were supported by quotes from the participants. Transferability was enhanced by describing the study context, and process for data collection and analysis.

Despite the clear evidence on the emerging themes related to self-care management education for heart failure patients, this study had some limitations. The data collection was time-consuming since this process delved into personal interaction during interviews, and discussions deviated from the main issue studied. Also because qualitative research is open-ended, participants had more control over the content of the data collected. We recognize the limitation of not exploring the views of other health workers who are involved directly in patient care. Their voices are important and might have added different perspectives. The insights gained from this study might provide insight into self-care management education in other hospitals given the similarities of how the other hospitals in Tanzania are organized.

## Conclusion

Poor self-care management for patients with heart failure results in readmission and prolonged hospital stay. Family involvement and the use of peer educators are the key steps in the improvement of self-care management for patients with HF. However, information overload and poverty that contribute to low uptakes of self-care management should be taken into consideration when planning for discharge for patients with HF. Self-care management education should be part of routine healthcare. Further research exploring the views of other health professional groups for heart failure patients would be helpful.

## Supporting information

**S1 Table. Dataset: Social demographic characteristics for the participants.**
(DOCX)

## Acknowledgments

Special appreciation is directed to the Jakaya Kikwete Cardiac Institute management for permission and support for the period of my study. The authors are also grateful to all JKCI team (staff) for their support and input.

## Author Contributions

**Conceptualization:** Peter M. Shirima.

**Data curation:** Peter M. Shirima.

**Formal analysis:** Peter M. Shirima.

**Methodology:** Peter M. Shirima.

**Resources:** Peter M. Shirima.

**Supervision:** Dickson A. Mkoka, Masunga K. Iseselo.

**Validation:** Peter M. Shirima.

**Writing – original draft:** Peter M. Shirima.

**Writing – review & editing:** Peter M. Shirima.

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
