## [Decision Letter · Decision Letter 0]

12 Mar 2024

PONE-D-23-35648Nurses’ perspectives on provision of self-care management education to patients with heart failure. A qualitative study at Cardiac hospital, Dar es Salaam Tanzania.PLOS ONE

Dear Dr. Shirima,

Thank you for submitting your manuscript to PLOS ONE. After careful consideration, we feel that it has merit but does not fully meet PLOS ONE’s publication criteria as it currently stands. Therefore, we invite you to submit a revised version of the manuscript that addresses the points raised during the review process.

We look forward to receiving your revised manuscript.

Kind regards,

Joyce Jebet Cheptum

Academic Editor

PLOS ONE

Journal Requirements:

2. Thank you for stating the following financial disclosure: "This research receive limited fund support from Muhimbili University of Health and Allied Sciences under a Project named East Africa Centre of excellence for Cardiovascular Sciences."  

3. Thank you for stating the following in the Acknowledgments Section of your manuscript: "We would like to thank the Muhimbili University of Health and Allied Sciences under the project named Centre of Excellency for Cardiovascular Sciences for supporting this study. "

"This research receive limited fund support from Muhimbili University of Health and Allied Sciences under a Project named East Africa Centre of excellence for Cardiovascular Sciences." 

Additional Editor Comments:

Title: Use colon instead of fullstop between the two sentences

Ln 30 - 33 - Rephrase the sentence, it is long and incomplete

On the abstract, add recommendations to Conclusion on the heading

Ln 105 Correct day to day to day

Ln 155 Delete However

Ln 172 The themes presented on the table are different from those listed in Ln 168 - 172. Clarify

Ln 297 Check spelling of provision

Ln 302 Edit Comma

Ln 307 - Ln 319 Compare the findings with other studies

Ln 363 - Jean et al., - Add the year of publication

Correct grammatical errors in the entire document.

Reviewers' comments:

Reviewer's Responses to Questions

**Comments to the Author**

1. Is the manuscript technically sound, and do the data support the conclusions?

Reviewer #1: Yes

Reviewer #2: Yes

2. Has the statistical analysis been performed appropriately and rigorously? 

Reviewer #1: N/A

Reviewer #2: Yes

3. Have the authors made all data underlying the findings in their manuscript fully available?

Reviewer #1: Yes

Reviewer #2: Yes

4. Is the manuscript presented in an intelligible fashion and written in standard English?

Reviewer #1: Yes

Reviewer #2: Yes

5. Review Comments to the Author

Reviewer #1: the research presents a technically sound qualitative study. It employs rigorous methods such as in-depth interviews and thematic analysis, appropriate for exploring the subjective experiences and perspectives of nurses. The manuscript's findings are well-supported by the data, adhering to the qualitative research paradigm. However, since it's a qualitative study, the concept of statistical analysis does not directly apply, as the study's strength lies in its thematic insights rather than quantifiable metrics.

The manuscript is well-written in standard English, facilitating clear understanding of its objectives, methods, findings, and conclusions. It follows a logical structure, enhancing readability and comprehension.

It's crucial to note that the manuscript should include a Data Availability Statement to comply with the PLOS Data policy. This statement is essential to verify that all underlying data supporting the findings are fully available without restriction or appropriately noted if any restrictions apply.

it contributes valuable insights into the challenges and facilitators of providing self-care management education to heart failure patients from the perspective of nurses in Tanzania. It highlights the importance of addressing barriers and leveraging facilitators to enhance patient care. Further attention to data availability and minor language corrections will strengthen the manuscript's submission readiness.

Reviewer #2: This manuscript is conducted in nice way written coherently n good English and it will be very helpful because it is can solve problems of the patients related with Heart failure. The development themes in the analysis part is also nice. All data is included.

6. PLOS authors have the option to publish the peer review history of their article (what does this mean?). If published, this will include your full peer review and any attached files.

Reviewer #1: **Yes: **Mohammed I. Al Bazroun

Reviewer #2: **Yes: **Michael Geletu Alaro

---

## [Author Response · Author response to Decision Letter 0]

5 Jun 2024

Reviewer #1: all comments provided are incorporated including grammar corrections. Thanks for your time to pass through this article.

Reviewer # 2: All comments provided are incorporated including grammar corrections. Thanks for your time to pass through this article.

---

## [Editor Report · Decision Letter 1]

19 Jun 2024

Nurses’ perspectives on provision of self-care management education to patients with heart failure. A qualitative study at Cardiac hospital, Dar es Salaam Tanzania.

PONE-D-23-35648R1

Dear Dr. Shirima,

We’re pleased to inform you that your manuscript has been judged scientifically suitable for publication and will be formally accepted for publication once it meets all outstanding technical requirements.

Kind regards,

Joyce Jebet Cheptum

Academic Editor

PLOS ONE
---

## [Editor Report · Acceptance letter]

3 Jul 2024

PONE-D-23-35648R1 

PLOS ONE

Dear Dr. Shirima, 

I'm pleased to inform you that your manuscript has been deemed suitable for publication in PLOS ONE. Congratulations! Your manuscript is now being handed over to our production team.

Kind regards, 

on behalf of

Dr. Joyce Jebet Cheptum 

Academic Editor

PLOS ONE